# Interface Formation of Medium-Thick AA6061 Al/AZ31B Mg Dissimilar Submerged Friction Stir Welding Joints

**DOI:** 10.3390/ma15165520

**Published:** 2022-08-11

**Authors:** Tifang Huang, Zeyu Zhang, Jinglin Liu, Sihao Chen, Yuming Xie, Xiangchen Meng, Yongxian Huang, Long Wan

**Affiliations:** 1State Key Laboratory of Advanced Welding and Joining, Harbin Institute of Technology, Harbin 150001, China; 2Harbin World Wide Welding Co., Ltd., Harbin 150001, China; 3Kunshan World Wide Welding Co., Ltd., Suzhou 215300, China; 4Shanghai Spaceflight Precision Machinery Institute, Shanghai 201600, China

**Keywords:** dissimilar metals, submerged friction stir welding, intermetallic compounds, mechanical properties

## Abstract

The medium-thick Al/Mg dissimilar friction stir welding (FSW) joint has serious groove and cavity defects due to uneven thermal distribution in the thickness direction. The submerged friction stir welding (SFSW) was employed to decrease the peak temperature of the joint and control the thermal gradient of the thickness direction, which were beneficial in suppressing the coarsening of the intermetallic compounds (IMCs) layer and improving the weld formation. According to the SEM results, the thickness value of the IMC layer in the nugget zone and shoulder affect zone decreased from 0.78 μm and 1.31 μm in FSW process to 0.59 μm and 1.21 μm in SFSW process at the same parameter, respectively. Compared with the FSW process, SFSW improves the thermal accumulation during the process, which inhibits the formation of the IMCs and facilitates the material flow to form a mechanical interlocking structure. This firm interface formation elevates the effective contact area of the whole joint interface and provides a strong connection between the dissimilar metals. Thus, the ultimate strength of the 6 mm thick Al/Mg dissimilar SFSW joints was enhanced to 171 MPa, equivalent to 71.3% of AZ31B Mg alloys strength.

## 1. Introduction

AZ31B Mg alloy is a wrought magnesium alloy with electromagnetic shielding performance. However, using only AZ31B Mg alloy will not only affect the signal transmission but also make the material susceptible to deformation and breakage in the process of use. 6061 Al alloy is a precipitation-hardened aluminum alloy which has great mechanical properties, corrosion resistance, and weldability. Thus, 6061 Al alloy and AZ31B Mg alloy hybrid structures have great potential applications in computer cases, communication, and consumer electronics due to the combination of lightweightness and high strength of Al and excellent electromagnetic shielding performance of Mg [1,2,3,4]. However, the considerable difference in physicochemical properties contributes to the difficulty of Al/Mg dissimilar welding [5]. As a promising solid-state welding technology, friction stir welding (FSW) has favorable advantages such as low thermal cycle and high-quality joints in connecting Al alloys, Mg alloys, and their dissimilar alloys [6]. Niu et al. [7] joined 7075 Al and AZ31 Mg alloys via FSW and detected Al_12_Mg_17_ and Al_3_Mg_2_ intermetallic compounds (IMCs) at the interface of the joint. The interface formation and material flow were the critical factors for the formation and performance of the Al/Mg dissimilar FSW joints [8]. Although the thermal cycle of FSW is much lower than fusion welding, the peak temperature of the FSW process can reach the eutectic reaction conditions of Al and Mg, leading to the formation of the brittle IMCs layer in the interfacial region [9]. Xu et al. [10] concluded that the weld formation and mechanical properties of the Al/Mg dissimilar joints deteriorated significantly with the increase of joint thickness. The ultimate strength of 3 mm thick 6061 Al and AZ31B Mg dissimilar FSW joint was about 175 MPa [8]. However, the maximum value of 6-mm thick dissimilar FSW joint was only 73 MPa [10]. The medium-thick or thick plates required adequate thermal cycle to guarantee the material flow in the root, which led to the overheat and IMCs coarsening in the upper region of the weld. The considerable difference in the microstructure of thickness direction resulted in welding defects such as groove, galling, and cavity.

Inhibiting the precipitation behavior of IMCs in the thickness direction of the weld is a significant challenge for obtaining stable dissimilar joints. Shi et al. [11] attempted to control the process parameters such as rotational velocity, welding speed, base metal positioning, and tool offset to reduce the IMCs. The interlayer materials such as Cu and Ni were added into the friction stir lap welded joints of Mg alloy and Al alloy to prevent the generation of Al-Mg IMCs [12]. Nevertheless, the influence of process parameters on heat input in the thickness direction was relatively limited, and the addition of interlayer also resulted in the generation of other IMCs.

Submerged friction stir welding (SFSW), which immerses the workpieces in a liquid environment, has been demonstrated to reduce the heat input and effectively limit the IMCs generation [13]. Zhang et al. [14] designed a water-cooling system to join AA6061 Al and pure copper and found that the SFSW joints possessed better formation and mechanical properties than the FSW joint. The temperature field model proved that the peak temperature of SFSW was dramatically lower than the FSW process [15]. The thickness of the IMCs decreased dramatically under the effect of water forced cooling. In addition, the dissimilar metals got fully mixed in the SFSW process and formed a mechanical interlocking structure, which was beneficial for the mechanical properties of the joints [16]. The previous studies have mainly focused on the microstructure and mechanical properties of thin plates dissimilar joints (<5 mm). The weld formation and the underlying relationship between the interface formation and cooling conditions of medium-thick Al/Mg dissimilar joints have not been well established.

In this paper, SFSW was proposed on 6 mm medium-thick AA6061 Al/AZ31B Mg dissimilar joints to control the thermal gradient in the thickness direction, reduce the IMCs layer, and improve the weld formation. The exclusive comparisons of microstructure evolution, interface formation, and mechanical properties of SFSW and FSW joints were comprehensively presented in this investigation.

## 2. Materials and Methods

AA6061-T6 Al alloy and AZ31B-O Mg alloy rolled plates with a thickness of 6 mm were used as base metals (BM) for dissimilar joining by SFSW. The nominal chemical compositions of the alloys are listed in Table 1. Gantry-type FSW equipment WWW-LM3324-2D-13T was used for SFSW and FSW experiments. The diameters of the shoulder, pin bottom and top of the rotational tool were 10 mm, 5 mm, and 3 mm. The tilt angle and plunge depth of the tool was 2.5° and 0.3 mm, respectively. Figure 1 displays the schematic of the cooling system and welding fixtures of SFSW. AZ31B Mg was placed at the advancing side (AS), and AA6061 was placed at the retreating side (RS). The offset value of the tool to the Mg side is 1.1 mm. The dissimilar SFSW joints were fabricated underwater and processed at the rotational velocity of 800 rpm, 900 rpm, and 1000 rpm with a constant welding speed of 70 mm/min. The cooling system kept the water temperature at 5 °C. For comparison, the FSW joint was welded at ambient temperature and the parameter was 800 rpm and 70 mm/min. 

After the welding process, the cross-section specimens were cut perpendicular to the welding direction and observed by optical microscopy (OM). X-ray diffraction (XRD) and scanning electron microscope (SEM) equipped with energy dispersive X-ray spectroscopy (EDS) were used to characterize the IMCs in the joint. Vickers microhardness measurements were carried out along the top, middle, and bottom of the cross-section with a load of 200 g and dwell time of 10 s. The horizontal and vertical interval of each point was 0.5 mm and 1.5 mm, respectively. Each point was measured three times. Three standard tensile samples were cut perpendicular to the weld for each joint to perform the tensile tests at a constant strain rate of 5 × 10^−4^ s. The schematic diagrams of microhardness and tensile samples based on the Chinese Standard GB/T2651 are shown in Figure 2.

## 3. Results

### 3.1. Macrostructures of the Joints

Figure 3 presents the surface morphologies of AA6061 Al/AZ31B Mg dissimilar joints fabricated via FSW and SFSW. As shown in Figure 3a,b, obvious galling and groove defects could be observed on the FSW joint while SFSW obtained a smooth weld surface at the same parameters. It indicated that the poor material flow in the FSW was greatly improved by the reduced heat input and rapid cooling rate in SFSW. As the rotational velocity increased, SFSW joints kept intact weld formation (Figure 3c,d).

Figure 4 displays the cross-section macrographs of the AA6061 Al/AZ31B Mg dissimilar FSW and SFSW joints. The stir zone (SZ) in each joint was immediately detectable. However, the different welding processes presented quite diverse mixing conditions. Figure 4a depicts the FSW joint fabricated at 800 rpm. It could be seen that severe cavity defect originated from the RS in the FSW weld, which was destructive to the joint performance. While for the SFSW-fabricated range 800 rpm to 1000 rpm, no interior defects such as cavity, void, and tunnel were observed in Figure 4b,d, which was the essential precondition for achieving the high-performance joint. The multiple material flow contributed to a complex mixing zone, including an upper zone owing to the predominant flow of the RS, and a violent vortex zone where both alloys completely penetrate each other. Small fragments and particles were dispersed at the whole vortex zone. These irregular-shaped scattered fragments provided a mechanical interlocking structure which provided more effective contact area of the interface and strengthened the interface bonding. As the rotational velocity increased, the mixing and dispersion of the BM in the SZ increased gradually. When the rotational velocity reached 1000 rpm, the excessive frictional heat enlarged the upper zone, which reduced the vortex zone and the effective mixing area of the interface, weakening the mechanical interlocking effect.

### 3.2. Microstructures of the Joints

To further investigate the microstructure evolution in FSW and SFSW joints, SEM images of the yellow rectangle zone in Figure 4a,b and the corresponding EDS mapping are shown in Figure 5. For the FSW joint fabricated at 800 rpm, partial Al alloy was stirred into the Mg matrix and microcavity defect marked with white arrow emerged in the interface (Figure 5a). During the mixing process, stress concentration arising at the brittle IMCs and formed the defect. Compared to the serious cavity in the upper and middle region, the IMCs layer may be relatively thinner owing to being far from the heat source. Thus, microcavity was formed in the lower region of the FSW joint. These microcavity provided the initiation sites and propagation paths of the crack, facilitating the failure of the joint. Differently, SFSW joint fabricated at the same rotational velocity presented a laminated structure including dark background, light lamellae, and gray bands phases (Figure 5d). The chemical compositions of the above phases were determined by EDS (Table 2). The results illustrated that the dark background (point 1 in Figure 5d) was the Mg alloy. The gray bands (point 2 in Figure 5d) composing both Al and Mg elements (much higher Mg content than Al) were Mg alloy and Al_12_Mg_17_. The light lamellae (point 3 in Figure 5d) were deformed Al pieces and Al_3_Mg_2_. It is consistent with the reports by Hamed et al. [17]. 

Besides the degree of intermixing structure, the interface between Al and Mg, especially in associations with the thickness of the IMCs layer, is particularly problematic for the dissimilar joint. Detailed SEM observations with EDS analysis of the interface region are shown in Figure 6. A gray diffusion interlayer distinguished the interface in the FSW weld between the Mg alloy (left side) and the Al alloy (right side). The EDS line analysis (Figure 6b) showed that the curve of Al and Mg element distribution processed a platform stage rather than continuously changing at the Al/Mg interface, revealing a metallurgical bonding at the interface. The IMCs near the Mg side was Al_12_Mg_17_. The IMCs near the Al side was Al_3_Mg_2_. However, the curve of element concentration of the Al/Mg interface in SFSW joint was much more distinct than in FSW joint (Figure 6c,d), indicating that the growth of IMCs was limited in the SFSW process. With the reduction of IMCs, the Al alloy and Mg alloy interpenetrate with each other at a higher degree and have a larger mutual penetration depth in SFSW joint than FSW joint. It facilitates better periodic mixing and stacking between the dissimilar alloys, resulting in a laminated structure.

Figure 7 presents the SEM images of IMCs in the AA6061 Al/AZ31B Mg interface in FSW and SFSW joints. As shown in Figure 7a,b, the IMCs thickness of SZ in FSW and SFSW joints were 0.78 μm and 0.59 μm, respectively. The submerged condition significantly dispersed the heat accumulation and reduced the dwelling time of high temperature, thus suppressing the growth of IMCs. Though the IMCs were continuously distributed at the Al/Mg interface, the IMCs layer thickness was changed owing to the different thermo-mechanical conditions. It should be emphasized that the IMCs layer thickness in the shoulder affect zone (SAZ) was higher than that in the SZ in both joints, as shown in Figure 7c,d. The IMCs thickness of SAZ in the FSW and SFSW joint were 1.31 μm and 1.21 μm due to SAZ probably suffering more frictional heat from the tool shoulder than SZ. 

To further verify the effect of temperature gradient on the IMCs layer, thermocouples were used to measure the thermal cycle of the thickness direction in FSW and SFSW joints. The thermocouples were located at three points in the heat-affected zone to avoid being broken by the rotational tool (Figure 8). As shown in Figure 9, the point T1 in the upper of the FSW weld possessed the maximum peak temperature (262 °C) due to proximity to the frictional heat source of the shoulder. The peak temperature of points T2 and T3 in the middle and bottom were 212 °C and 206 °C. For the SFSW joint, the peak temperature of the three points was dropped to 128 °C, 106 °C, and 56 °C, respectively. It indicated that SFSW could effectively reduce the temperature of the upper surface of the weld and thermal gradient in the thickness direction, keeping the homogeneous microstructure.

## 4. Discussion

### 4.1. Formation Mechanism of IMCs

Sharifitaba et al. [9] believed that the eutectic reaction and solid-state diffusion are the dominant reason for the generation of IMCs. The friction heat during the FSW process brings the weld to the eutectic condition, resulting in the local melting and interfacial liquid film emerging along the grain boundary. It promotes the rapid diffusion of Al and Mg atoms until solidification. The eutectic reactions of Al–Mg binary phases are shown as follows [18]:427 °C: L→Al_12_Mg_17_ + Mg,(1)
450 °C: L→Al_3_Mg_2_ + Al,(2)

To further explore the underlying interplay between welding temperature and the formation of IMCs, the reaction kinetics of Al_12_Mg_17_ and Al_3_Mg_2_ were calculated and shown in Figure 10. It could be seen that the Gibbs energy of IMCs was inversely proportional to the temperature. Thus, it is difficult for IMC to nucleate at a lower temperature. The Gibbs energy of Al_12_Mg_17_ was lower than Al_3_Mg_2_. Hence, it demonstrated that the Al_12_Mg_17_ phase appeared foremost and accumulated at the Al-Mg interface near the Mg side.

According to the observations, thermodynamics and kinetics discussion, the mechanisms of IMCs formation at the Al-Mg interface are shown in Figure 11. First, the initial butted interface inevitably had a slight assembly gap (Figure 11a). With the severe plastic deformation of FSW, sufficient material flow immediately mixed the Al and Mg matrices. The continued thermal cycle promoted the diffusion of the Al and Mg atoms. As the welding temperature reached the eutectic condition of Al_12_Mg_17_ and Al_3_Mg_2_, the interfacial liquid film was formed (Figure 11b). When the welding temperature of the SFSW process was kept below the eutectic temperatures, it was difficult for the interface to generate the liquid film. Thus, IMCs nucleated and grew just via interdiffusion between the alloys, which was remarkably slower than the FSW process. At the interface near the Mg side, Al_12_Mg_17_ with a lower eutectic temperature and the Gibbs energy preferentially nucleated (Figure 11c). With the continuous interdiffusion near the interface, Mg atoms accumulated on the Al side and formed lamellar Al_3_Mg_2_ (Figure 11d). The forced water-cooling effect of SFSW effectively reduces the heat accumulation, especially in the upper region of the weld, increasing the difficulty in the atom diffusion and IMC reaction simultaneously. Therefore, SFSW is beneficial in inhibiting the growth of IMCs and controlling the weld formation in the thickness direction. It also explains why the IMCs layer of SAZ is thicker than that of SZ in both FSW and SFSW joints, as the SAZ is subject to more frictional heat.

### 4.2. Microhardness Distribution

The microhardness distribution profiles of the SFSW dissimilar joint were measured along the dashed lines marked in the inset of Figure 12. The error bar fluctuated within 4 HV, representing the microhardness result of each point was reliable. The average hardness values of 6061 Al alloy and AZ31B Mg alloys were 110 HV and 61 HV, respectively. The hardness of SZ boundary in both sides was lower than that of BM because of the grain coarsening and precipitates dissolution caused by the heat accumulation. The minimum value (51 HV) of the dissimilar joint SFSW was located at the Mg side of SZ boundary. However, an inhomogeneous distribution of hardness values was observed in the SZ. The hardness value at the middle and bottom was much higher than that at the upper SAZ due to the great mixing degree in the vortex zone.

### 4.3. Tensile Properties and Fracture Analysis

Due to FSW joint fabricated at 800 rpm having severe cavity defects, tensile tests were carried out only on well-formed SFSW joints. Figure 13 shows the tensile test results of SFSW dissimilar joints processed at 800 rpm, 900 rpm, and 1000 rpm. Owing to the poor weld formation of the FSW joint, tensile test was not performed on FSW joint. The ultimate strength of the SFSW joint processed at 800 rpm fluctuated slightly around 171 MPa, equivalent to 71.3% of AZ31B Mg alloy strength. With the increase of rotational velocity, the ultimate strength gradually decreased. When the rotational velocity reached 1000 rpm, the joint possessed the lowest ultimate strength, only 135 MPa. It could be attributed to the decrease of interface effective contact area and the thicker IMCs layer. In the submerged condition, the transient thermal cycle of high rotational velocity led to the coarsening of IMCs, which was deleterious to the mechanical property.

Figure 14 illustrates the fracture paths in the SFSW joints. The fracture position of SFSW joints at 800 rpm and 1000 rpm were located at the RS of SZ boundary. Due to the lack of sufficient mixing degree, the crack first formed at the unmixing upper zone and then propagated along the no mechanical interlocking SZ boundary rapidly. It demonstrated that the intermixing structure in SZ impeded crack propagation and exerted an excellent mechanical interlocking effect. The SFSW joint fabricated at 900 rpm fractured at AS of the SZ boundary, where the lowest hardness was located and mechanical interlocking was absent.

Figure 15 displays the tensile fractured surface of SFSW dissimilar joints. Apparent tear ridges and fine dimples were observed on the fracture surface of SFSW at 800 rpm, indicating a combined ductile-brittle fracture. When the rotational velocity was 900 rpm, the fracture surface has smooth cleavage surface and a small number of dimples. However, there were obvious tear ridges on the 1000 rpm joint fracture surfaces, indicating brittle fracture. The large amounts of IMCs generated in high rotational velocity led to an increase in brittleness of the joint. Some dispersed particles could be observed at the bottom of the dimples. Figure 16 shows the XRD result of the fracture surface. In addition to Al and Mg alloy, it univocally identifies that the dispersed particles present on the fracture surface were Al_12_Mg_17_ and Al_3_Mg_2_ phase, which were the crack initiation and propagation. 

## 5. Conclusions

Sound medium-thick 6061Al/AZ31B Mg dissimilar joint was fabricated by SFSW. The effect of the cooling system on the thermal cycle in the thickness direction and microstructural evolutions was discussed. The most important findings are summarized below:(1)Compared with FSW joint, no groove, galling, and cavity defects were detected in SFSW joints. The ultimate strength of the dissimilar SFSW joints was enhanced to 171 MPa, equivalent to 71.3% of AZ31B Mg alloys strength.(2)Thermodynamics and kinetics analysis illustrate the interface formation mechanism. The improved thermal cycle and temperature gradient in SFSW limited the nucleate and growth of the IMCs in the thickness direction of the weld. The IMC layer thickness in the nugget zone and shoulder affect zone was reduced from 0.78 μm and 1.31 μm in the FSW process to 0.59 μm and 1.21 μm in the SFSW process.(3)The eutectic reactions have been suppressed by the SFSW process, which is favorable to limiting the generation of brittle IMCs and improving the plastic material flow. Thus, the complex laminated structure in SFSW weld exerts mechanical interlocking of the dissimilar metals, strengthening the interface connection.

## Figures and Tables

**Figure 1 materials-15-05520-f001:**
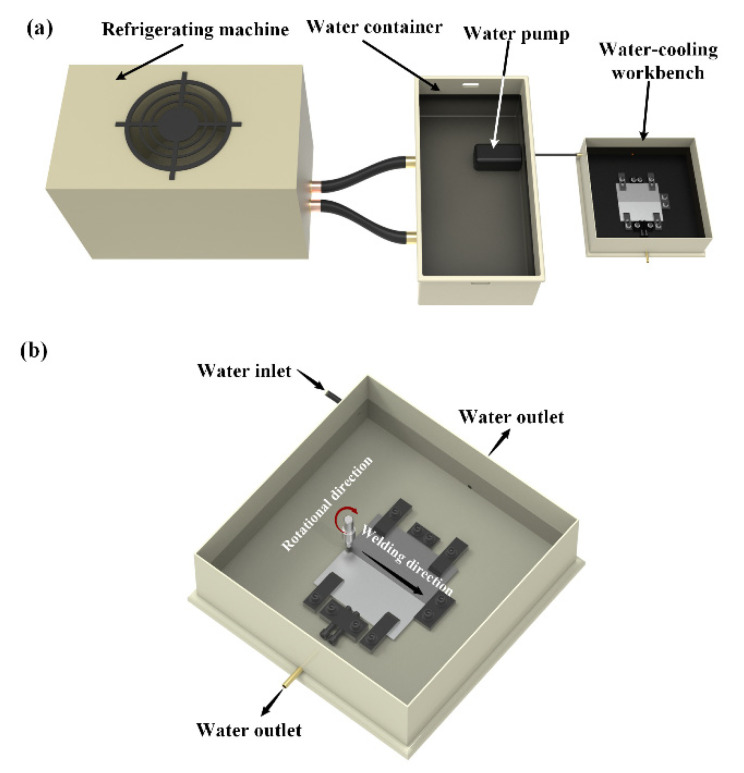
Schematics of the SFSW: (**a**) cooling system, (**b**) welding fixture.

**Figure 2 materials-15-05520-f002:**
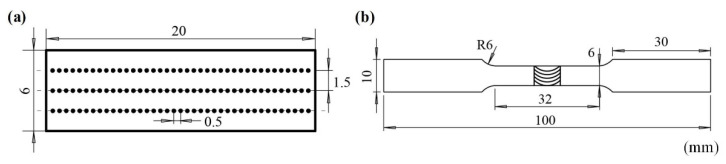
Schematics of (**a**) microhardness test sample and (**b**) tensile test sample.

**Figure 3 materials-15-05520-f003:**
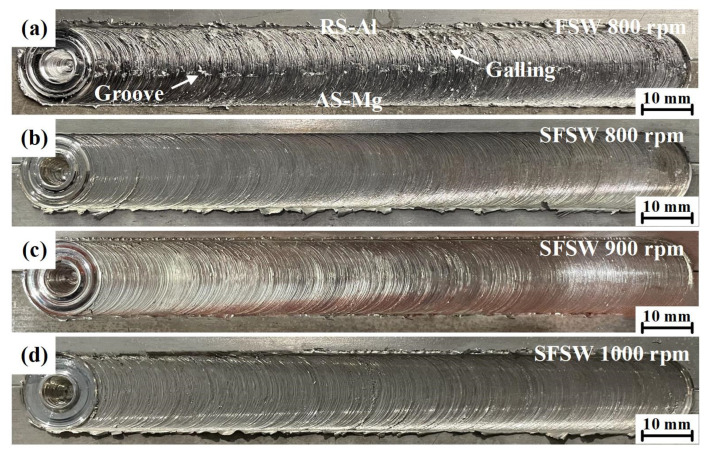
Surface morphologies of AA6061 Al/AZ31B Mg dissimilar joints produced by (**a**) FSW at 800 rpm, (**b**–**d**) SFSW at 800 rpm, 900 rpm and 1000 rpm.

**Figure 4 materials-15-05520-f004:**
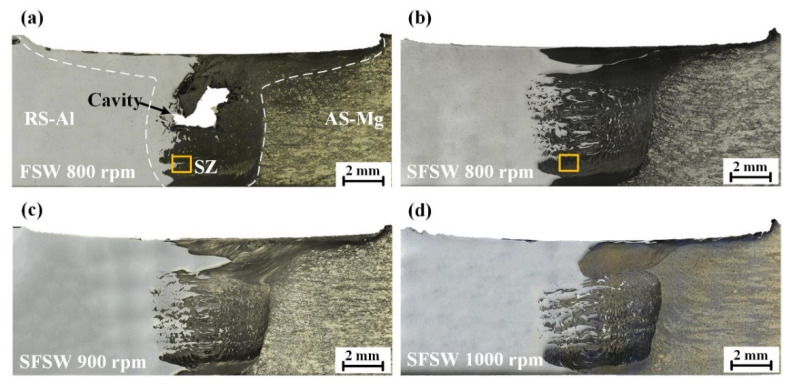
Cross-section macrographs of the joints produced by (**a**) FSW at 800 rpm, (**b**–**d**) SFSW at 800 rpm, 900 rpm, and 1000 rpm.

**Figure 5 materials-15-05520-f005:**
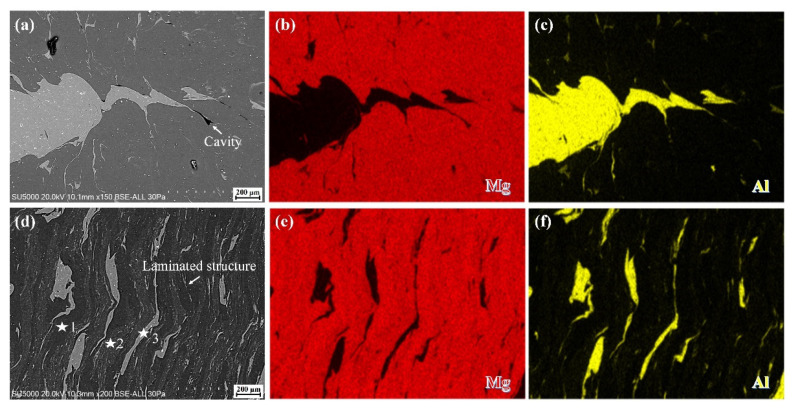
SEM images and EDS mapping of the microstructure in (**a**–**c**) FSW and (**d**–**f**) SFSW weld fabricated at 800 rpm.

**Figure 6 materials-15-05520-f006:**
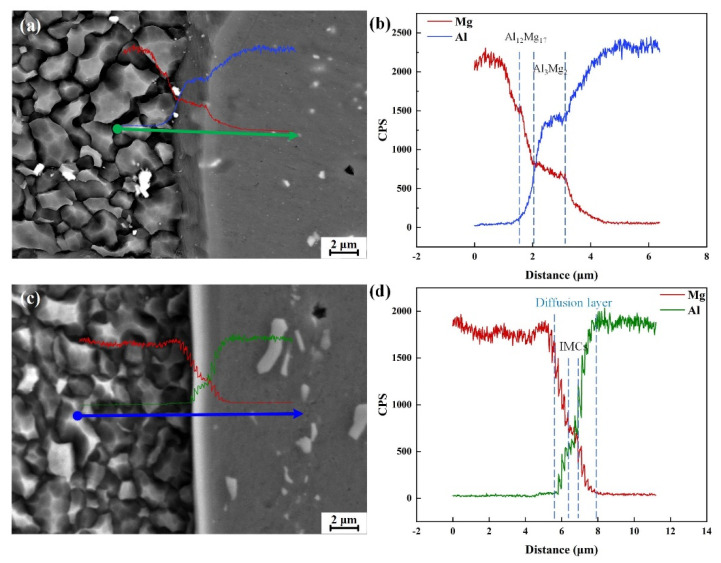
SEM and EDS results of the interface between Al and Mg in the (**a**,**b**) FSW and (**c**,**d**) SFSW weld fabricated at 800 rpm.

**Figure 7 materials-15-05520-f007:**
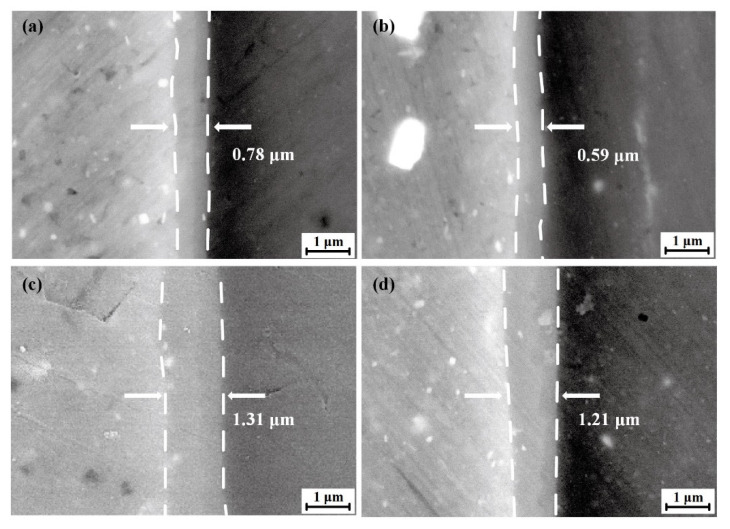
SEM images of IMCs at different conditions (**a**) SZ of FSW weld, (**b**) SZ of SFSW weld, (**c**) SAZ of FSW weld, (**d**) SAZ of SFSW weld.

**Figure 8 materials-15-05520-f008:**
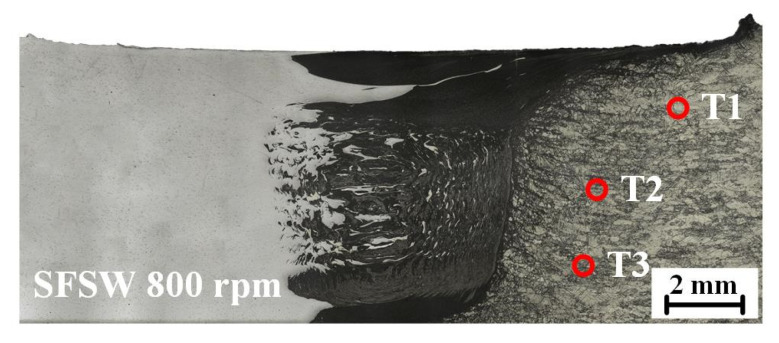
Diagram of the thermocouple position.

**Figure 9 materials-15-05520-f009:**
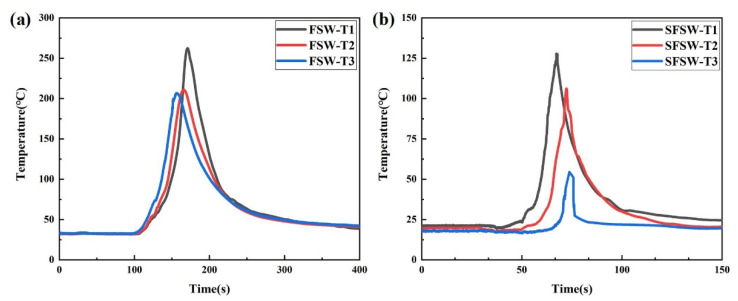
Measured thermal cycles in the (**a**) FSW and (**b**) SFSW.

**Figure 10 materials-15-05520-f010:**
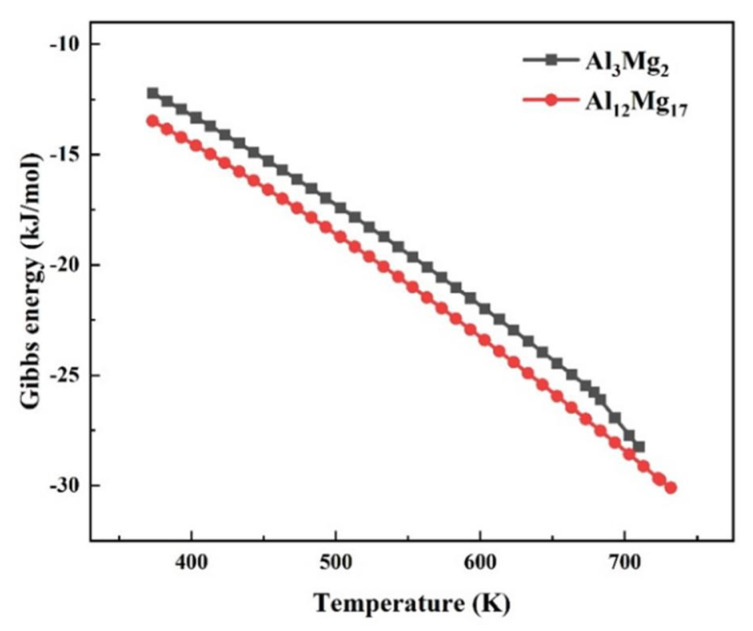
The Gibbs energy curve of the IMCs.

**Figure 11 materials-15-05520-f011:**
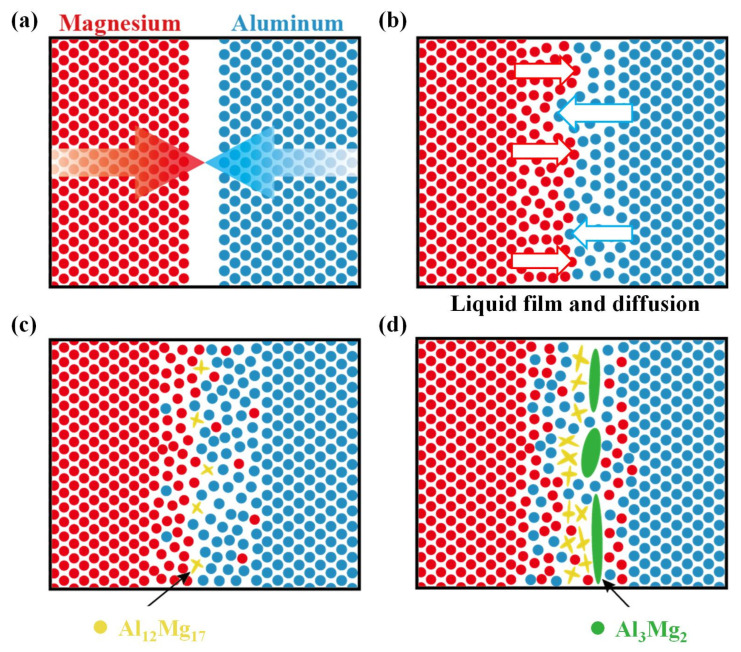
Schematic diagrams for the formation mechanism of IMCs: (**a**) the original interface, (**b**) liquid film and Al-Mg diffusion, (**c**) Al_12_Mg_17_, (**d**) Al_3_Mg_2_.

**Figure 12 materials-15-05520-f012:**
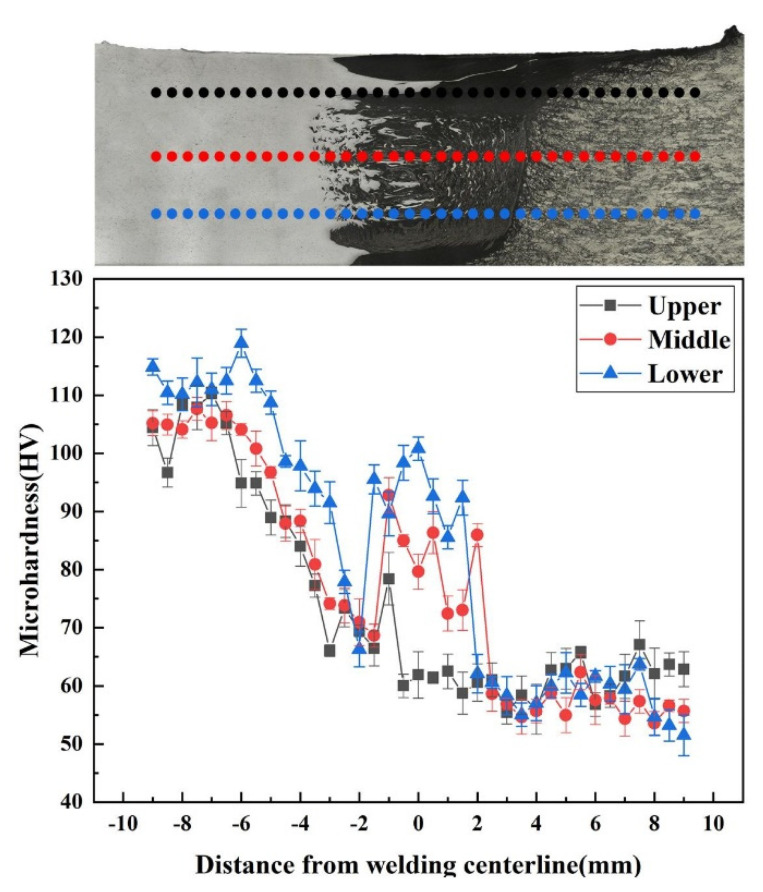
Microhardness distribution profile of the SFSW dissimilar joint.

**Figure 13 materials-15-05520-f013:**
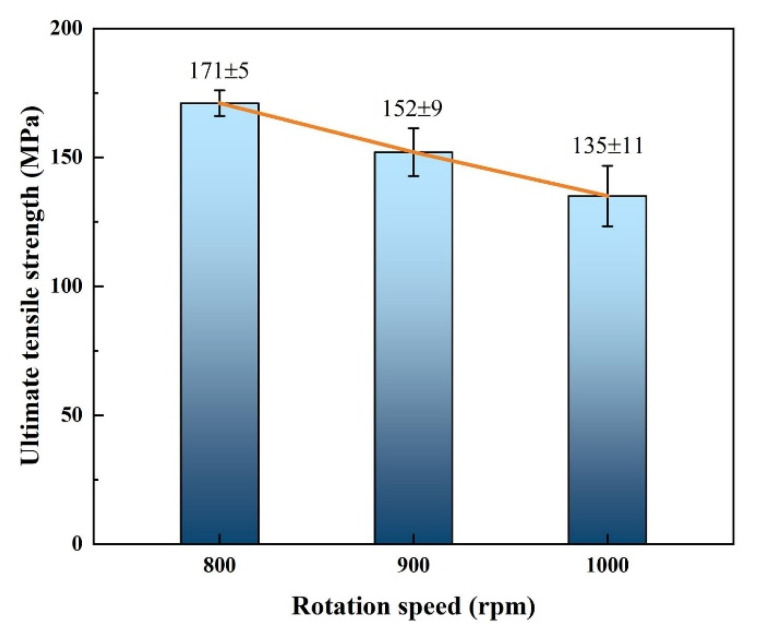
Ultimate tensile strength of the SFSW joints fabricated at different rotational velocities.

**Figure 14 materials-15-05520-f014:**
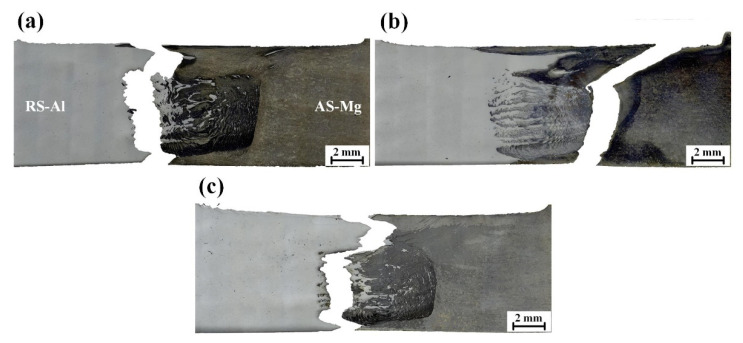
Fracture positions of SFSW at (**a**) 800 rpm, (**b**) 900 rpm, and (**c**) 1000 rpm.

**Figure 15 materials-15-05520-f015:**
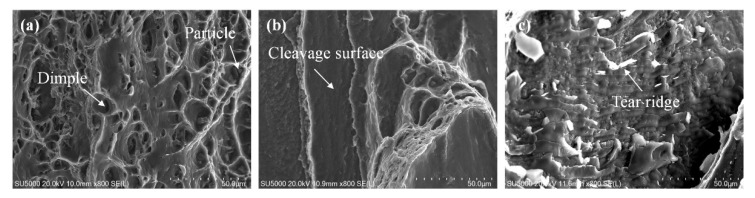
Fracture surfaces of SFSW at (**a**) 800 rpm, (**b**) 900 rpm, and (**c**) 1000 rpm.

**Figure 16 materials-15-05520-f016:**
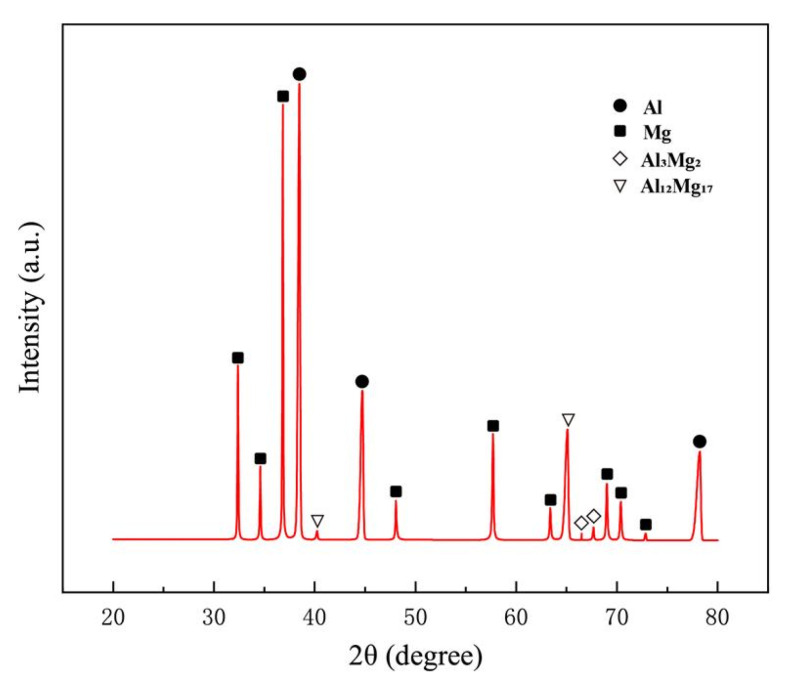
XRD results of the fracture surface.

**Table 1 materials-15-05520-t001:** Nominal chemical compositions of base metals.

Materials	Nominal Chemical Composition (wt.%)									
	Si	Fe	Cu	Mn	Mg	Cr	Zn	Ti	Al	Ni
6061-T6	0.4–0.8	0.7	0.15–0.4	0.15	0.8–1.2	0.1	0.25	0.15	Bal.	-
AZ31B	0.016	0.001	0.003	0.48	Bal.	-	0.88	-	3.1	0.0009

**Table 2 materials-15-05520-t002:** EDS results in the location marked in Figure 5.

Point	Al (at. %)	Mg (at. %)	Possible Composition
1	12.31	87.69	Mg
2	21.16	78.84	Mg + Al_12_Mg_17_
3	80.34	19.66	Al + Al_3_Mg_2_

## Data Availability

Not applicable.

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
