# Peer review of "Interface Formation of Medium-Thick AA6061 Al/AZ31B Mg Dissimilar Submerged Friction Stir Welding Joints"

_materials, 2022, doi:10.3390/ma15165520_

Round 1

Reviewer 1 Report

The manuscript is well organized. The discussion is quite interesting.

(1) Figure 7

The thicknesses of the IMC layers in Figs. 7(a) and 7(c) are thinner than that in Fig. 6(a). Why the thickness of the IMC layer has been changed?

(2) Discussion (3)

“The material flow was improved in the SFSW process due to the reduced IMCs.”

The improvement of the material flow during the SFSW process has not been clarified. Instead of that, the eutectic reactions has been suppressed by the SFSW process.

Author Response

1. The thicknesses of the IMC layers in Figs. 7(a) and 7(c) are thinner than that in Fig. 6(a). Why the thickness of the IMC layer has been changed?

RESPONDS:

Thank you for the reminding. We did a mistake in the scale bars in Fig 6 (a) and Fig 6 (c), which result in the thickness of the IMC layer in Fig 6 being larger than that in Fig 7.

We have revised it and added the right one in Fig 6.

2. “The material flow was improved in the SFSW process due to the reduced IMCs.” The improvement of the material flow during the SFSW process has not been clarified. Instead of that, the eutectic reactions has been suppressed by the SFSW process.

RESPONDS:

Thank you for the reminding. We used cross-section macrographs and SEM images to observe the mixing conditions of the joint interface and found that the SFSW joint possessed a much more complex mixing zone than the FSW joint due to the improvement of coarsening IMCs. By analyzing the formation mechanism of IMCs, we confirmed that the decrease in the thermal cycle inhibited the eutectic reaction and IMCs generation.

The eutectic reactions have been suppressed by the SFSW process, which is favorable to limiting the generation of brittle IMCs and improving the plastic material flow.

We have revised this part in 303rd line.

Reviewer 2 Report

The authors studied submerged friction stir welding of aluminum and magnesium alloys. The article is well structured, but it should be supplemented before publication in the journal. It is not clear why the research was carried out. The data obtained, especially the mechanical properties, are related to the microstructure, but the authors do not provide such studies. There are also comments on the text of the article:

1. What are these materials for? Why choose aluminum with magnesium alloys? References need to be expanded and highlight the purpose of the work.

2. How are base metals obtained and in what state are they used?

3. The equipment should be indicated in the article.

4. What is the difference between SFSW and FSW is not clear from the text.

5. It is necessary to clarify the procedure for obtaining samples, for example, for fig. 4.

6. Where is table 1 with nominal chemical compositions of the alloys?

7. How is Possible composition defined in Table 1. EDS results in the location marked?

8. What do the authors mean in line 156 by "metallurgical reaction"?

9. Why do the authors believe that the indicated in fig. 7 sites is IMC?

10. Which SFSW mode is used in fig. 5 and fig. 6?

11. Please, check reactions (1) and (2).

12. There are no microstructural studies in the article, which is important for this article.

13. It is necessary to indicate the error in the measurement of microhardness and justify changes.

14. In what mode were tests for mechanical properties carried out?

15. What is the difference between table 1 and 2? Where are the points 1,2,3 taken from?

Author Response

1. What are these materials for? Why choose aluminum with magnesium alloys? References need to be expanded and highlight the purpose of the work.

RESPONDS:

Thank you for the reminding. We did not express it clearly before.

6061 Al alloy and AZ31B Mg alloy hybrid structures have great potential applications in computer cases, communication and consumer electronics due to combining lightweight, high strength of Al and excellent electromagnetic shielding performance of Mg [1-3].

We have added this part in 32nd line in the Introduction section.

2. How are base metals obtained and in what state are they used?

RESPONDS:

Thank you for the reminding. As suggested by the reviewer, we supplemented the detail of the base metals.

AA6061-T6 Al alloy and AZ31B-O Mg alloy rolled plates with a thickness of 6 mm were used as base metals (BM) for dissimilar joining by SFSW.

We have added this part in the 80th line in the Introduction section.

3. The equipment should be indicated in the article.

RESPONDS:

Thank you for the recommendation.

Gantry-type FSW equipment WWW-LM3324-2D-13T was used for SFSW and FSW experiments.

As suggested, we have added this part in the 82nd line in Materials and Methods section.

4. What is the difference between SFSW and FSW is not clear from the text.

RESPONDS:

Thank you for the reminding. We did not explain the difference before.

Submerged friction stir welding (SFSW), which immerses the workpieces in a liquid environment, has been demonstrated to reduce the heat input and effectively limit the IMCs generation [12].

We have added these parts in the 61st line in Introduction section.

5. It is necessary to clarify the procedure for obtaining samples, for example, for fig. 4.

RESPONDS:

Thank you for the reminding. As suggested, we have clarified the procedure of the samples in the text.

Fig. 4a depicts the FSW joint fabricated at 800 rpm. It could be seen that severe cavity defect originated from the RS in the FSW weld, which was destructive to the joint performance. While for the SFSW fabricated range 800 rpm to 1000 rpm, no interior defects like cavity, void and tunnel were observed in Fig. 4b-d, which was the essential precondition for achieving the high-performance joint.

We have added these parts in the 120th line in Macrostructures of the joints section.

6. Where is table 1 with nominal chemical compositions of the alloys?

RESPONDS:

Thank you for the reminding. We have supplemented the table 1 with nominal chemical compositions of the base metals in the Materials and Methods section.

7. How is Possible composition defined in Table 1. EDS results in the location marked?

RESPONDS:

Thank you for the reminding. According to the EDS result, we could obtain the atomic ratio of the elements and calculate their weight ratios, and obtain the possible composition. It is essential to note that X-rays can be absorbed and therefore, quantification of elements using soft X-ray lines could lead to errors. Thus, we also did XRD to confirm the possible composition. The EDS point scanning was carried out on the points marked in Fig. 5d.

The chemical compositions of the above phases were determined by EDS (Table. 2). The results illustrated that the dark background (point 1 in Fig. 5d) was the Mg alloy. The gray bands (point 2 in Fig. 5d) composing both Al and Mg elements (much higher Mg content than Al) were Mg alloy and Al12Mg17. The light lamellae (point 3 in Fig. 5d) were deformed Al pieces and Al3Mg2. It is consistent with the reports by Hamed et al. [16].

We have added this part in 150th line.

8. What do the authors mean in line 156 by "metallurgical reaction"?

RESPONDS:

Thank you for the reminding. We did not express it clearly. It should be metallurgical bonding rather than metallurgical reaction. The IMCs layer at the Al-Mg interface could exert metallurgical bonding effect.

We have revised it in 165th line.

9. Why do the authors believe that the indicated in fig. 7 sites is IMC?

RESPONDS:

Thank you for the reminding. The EDS line scanning result of Fig. 6 proved that the interface between Al and Mg was composed of IMCs. Fig. 7 shows other interfaces at different conditions of the FSW and SFSW joint. Thus, we believed that the interface site in Fig.7 was the IMC layer.

10. Which SFSW mode is used in fig. 5 and fig. 6?

RESPONDS:

Thank you for the reminding. SFSW joint fabricated at 800 rpm was used in Fig. 5 and Fig. 6.

For the FSW joint fabricated at 800 rpm…Differently, SFSW joint fabricated at the same rotational velocity presented a laminated structure including dark background, light lamellae and gray bands phases

SEM images and EDS mapping of the microstructure in (a-c) FSW and (d-f) SFSW weld fabricated at 800 rpm.

We have added these parts in 141st line and figure title in Fig. 5 and Fig. 6.

11. Please, check reactions (1) and (2).

RESPONDS:

Thank you for the reminding. We did a mistake in reaction 1 and reaction 2.

427℃: L→Al12Mg17 + Mg,

450℃: L→Al3Mg2 + Al,

We have revised these parts in 210th line.

12. There are no microstructural studies in the article, which is important for this article.

RESPONDS:

Thank you for the reminding. Our research was focused on the IMCs coarsen and formation mechanism in medium-thick dissimilar SFSW joints. The microstructure evolution from FSW to SFSW joint was discussed in Fig. 5 and Fig. 6. The change of IMCs layer thickness in the thickness direction of the joints were analyzed in Fig. 7. The formation mechanism of IMCs layer and growth were evaluated from thermodynamics and kinetics in Fig. 9-11.

13. It is necessary to indicate the error in the measurement of microhardness and justify changes.

RESPONDS:

Thank you for the reminding. Generally, microhardness was tested along the cross-section of the joint because the FSW joint have different zone like nugget zone, thermo-mechanically affected zone, heat affect zone and base metal. Since the base metals are medium-thick plates, microhardness tests were carried out along the top, middle and bottom of the cross-section to analyze the difference. For FSW joint or other fusion weld joints, one point only need one test because the interval of each point was very small. In our research. The horizontal interval of each point was only 0.5 mm. It could reflect the microhardness and the trend of the change in joint. Thus, we did not test it more.

14. In what mode were tests for mechanical properties carried out?

RESPONDS:

Thank you for the reminding. We didn’t test the mechanical properties continually due to the FSW joint fabricated at 800 rpm had severe cavity defects. So, we carried out mechanical tests on SFSW joints fabricated at 800 rpm, 900 rpm and 1000 rpm.

Due to FSW joint fabricated at 800 rpm having severe cavity defects, so tensile tests were carried out only on well-formed SFSW joints. Fig. 13 shows the tensile test results of SFSW dissimilar joints processed at 800 rpm, 900 rpm and 1000 rpm.

We have added this part in 253rd line.

15. What is the difference between table 1 and 2? Where are the points 1,2,3 taken from?

RESPONDS:

Thank you for the reminding. The EDS result in table 1 was used to confirm the chemical compositions of the laminated structure, which provided the foundation for the subsequent discussion on the formation mechanism of IMCs. The EDS result in table 2 was used to validate the presence of IMCs on the fracture surface, which explained the reason of the crack initiation and propagation.

Reviewer 3 Report

Authors have conducted the experiments on AA6061 and Mg joints prepared by submerged and normal friction stir welding.

To improve the quality, it is required to incorporate following in the manuscript.

1. Include IMC acronym in abstract.

2. Reference 3 is not cited in the manuscript.

3. Specify the ASTM standards for microhardness and tensile tests in the manuscript.

4. Give some more details about XRD patterns.

5. Mark various fracture mechanisms in Fig. 15.

6. Explain some details about fracture behavior of tensile fractured surfaces.

Author Response

1. Include IMC acronym in abstract.

RESPONDS:

Thank you for the reminding. We have added the intermetallic compounds (IMCs) in 19th in the abstract section.

2. Reference 3 is not cited in the manuscript.

RESPONDS:

Thank you for the reminding. We have supplemented reference 3 in 34th line in the introduction section.

3. Specify the ASTM standards for microhardness and tensile tests in the manuscript.

Thank you for the reminding. We used the Chinese Standard GB/T2651 for microhardness and tensile tests.

The schematic diagrams of microhardness and tensile samples based on the Chinese Standard GB/T2651 are shown in Fig. 2.

We have added this part in the 100th line.

4. Give some more details about XRD patterns.

RESPONDS:

Thank you for the reminding.

Since EDS point scanning have been carried out on the particle, the XRD was used to assist in the confirmation of Al3Mg2 and Al12Mg17 on the fracture surface. Thus, we did not discuss more details about XRD.

5. Mark various fracture mechanisms in Fig. 15.

RESPONDS:

Thank you for the reminding. As suggested, we have marked the fracture mechanism in Fig. 15.

6. Explain some details about fracture behavior of tensile fractured surfaces.

RESPONDS:

Thank you for the reminding.

Apparent tear ridges and fine dimples were observed on the fracture surface of SFSW at 800 rpm, indicating a combined ductile-brittle fracture. When the rotational velocity was 900 rpm, the fracture surface has smooth cleavage surfaces and a small number of dimples. However, there were obvious tear ridges on the 1000 rpm joint fracture surfaces, indicating brittle fracture. The large amounts of IMCs generated in high rotational velocity led to an increase of the brittleness of the joint.

We have added this part in 276th line.

Reviewer 4 Report

This paper aims to study the Interface formation of medium-thick AA6061 Al/AZ31B Mg dissimilar submerged friction stir welding joints. Some corrections need to be made, for example:

1.     Please provide additional information about the mechanism of the formation of cavities and cracks in Fig. 5. More evidence is required to demonstrate this is due to the uneven thermal distribution in the thickness direction.

2.     Why does SFSW joint present a laminated structure?

3.     What effect does the grain size of the welding zone have on the different properties of the samples?

4.     The following paper showed that the segregation of the Mg-solute at grain boundaries increases the possibility of corrosion in the specimens and leads to a decrease in the mechanical properties. Authors should explain the effect of Mg segregation.  https://doi.org/10.1016/j.vacuum.2020.109937

Author Response

1. Please provide additional information about the mechanism of the formation of cavities and cracks in Fig. 5. More evidence is required to demonstrate this is due to the uneven thermal distribution in the thickness direction.

RESPONDS:

Thank you for the reminding. We did not express the above questions clearly before.

Fig. 5a shows the yellow rectangle zone in Fig. 4a. It is located in the lower region of the FSW joint. Only microcavity was observed at the interface. However, the upper and middle area of the joint has a severe cavity. According to the measured thermal cycle during the FSW process in Fig. 9, we found that the peak temperature was in the upper region due to proximity to the frictional heat source of the shoulder, leading to large amounts of IMCs which seriously hindered the material flow. The IMCs layer was not coarsening seriously at the lower region of the joint due to being far from the heat source, and the weld formation (microcavity) was relatively better compared to the top region. The SEM images of the IMC layer in different regions in the thickness direction could also prove this phenomenon. Thus, we concluded that uneven thermal distribution in the thickness direction was the reason for these defects of medium-thick Al/Mg dissimilar FSW joints.

During the mixing process, stress concentration arises at the brittle IMCs and formed the defect. Compared to the serious cavity in the upper and middle region, the IMCs layer may be relatively thinner owing to being far from heat source. Thus, microcavity was formed in the lower region of the FSW joint.

We have added this part in 143rd line.

2. Why does SFSW joint present a laminated structure?

RESPONDS:

Thank you for the reminding.

SFSW decreases the thermal cycle and inhibits the eutectic reaction. Thus, the IMCs layer is greatly limited, which could be proved by SEM results of the interface. With the reduction of IMCs, the Al and Mg materials interpenetrate with each other at a higher degree and have a larger mutual penetration depth in SFSW joint than FSW joint. It facilitates better periodic mixing and stacking between the dissimilar alloys, resulting in a laminated structure.

We have added this part in 169th line.

3. What effect does the grain size of the welding zone have on the different properties of the samples?

RESPONDS:

Thank you for the reminding. According to Hall-Petch relationship, the finer the grain and the higher the strength. SFSW could reduce the peak temperature and accelerate the cooling rate of the joint. In addition, the heat input is higher as the rotational velocity increases. Consequently, the SFSW joint fabricated at 800 rpm possesses more refined grains and better mechanical properties.

4. The following paper showed that the segregation of the Mg-solute at grain boundaries increases the possibility of corrosion in the specimens and leads to a decrease in the mechanical properties. Authors should explain the effect of Mg segregation. https://doi.org/10.1016/j.vacuum.2020.109937

RESPONDS:

Thank you for the reminding. Generally, segregation often occurs during fusion welding with a high diffusion rate. Friction stir welding is a solid-state welding technology. The joint is welded via frictional heat and severe plastic deformation which results in the limitation of element diffusion. SFSW immerses the workpieces in a liquid environment, which further reduces the heat input. Thus, reports related to FSW or SFSW rarely discuss segregation. But the mentioned paper provides important guidance for our following work, and we have added the reference in the introduction section.

Round 2

Reviewer 2 Report

The authors responded to several comments, but some questions remained unanswered.

1. Why choose such aluminum alloy and such magnesium alloy? Why in T6 state?

2. Figure 7 and its explanation is unclear. IMC is a phase, it cannot be a continuous layer in any way.

3. Microhardness measurement without indicated error cannot show the difference in values, there may be an error so large that there is no change in microhardness.

4. At what strain rate were tensile tests performed?

5. Possible compositions in Table 3 are unproven.

Author Response

The authors responded to several comments, but some questions remained unanswered.

  1. Why choose such aluminum alloy and such magnesium alloy? Why in T6 state?

RESPONDS:

Thank you for the reminding. We did not express it clearly. AZ31B Mg alloy is a wrought magnesium alloy with electromagnetic shielding performance. Thus, AZ31B is the preferred material for consumer electronics computer cases to reduce the risk of electromagnetic radiation. However, only using magnesium alloy will not only affect the signal transmission but also easy to deformation and breakage in the process of use. 6061 Al alloy is a precipitation-hardened aluminum alloy which has great mechanical properties and weldability. It is commonly available in pre-tempered such as solutioned and artificially aged (T6). T6 temper could provide the maximum precipitation hardening for a 6061 Al alloy. 6061 Al as the material of the computer case can ensure strength and corrosion resistance. Therefore, many manufacturers start to fabricate Al/Mg hybrid structures to achieve a balance between electromagnetic shielding performance and mechanical properties.

AZ31B Mg alloy is a wrought magnesium alloy with electromagnetic shielding performance. However, only using magnesium alloy will not only affect the signal transmission but also easy to deformation and breakage in the process of use. 6061 Al alloy is a precipitation-hardened aluminum alloy which has great mechanical properties, corrosion resistance and weldability. Thus, 6061 Al alloy and AZ31B Mg alloy hybrid structures have great potential applications in computer cases, communication and consumer electronics due to combining lightweight, high strength of Al and excellent electromagnetic shielding performance of Mg [1-3].

We have added this part in the 32nd line in the Introduction section.

  1. Figure 7 and its explanation is unclear. IMC is a phase, it cannot be a continuous layer in any way.

RESPONDS:

Thank you for the reminding. IMCs is a type of solid phase which is formed via two or more metallic elements. For dissimilar welding like FSW and diffusion welding, continuous IMCs layer is formed in the interface of the joint due to reactive diffusion. According to the paper “Shi H, Chen K, Liang Z, et al. Intermetallic compounds in the banded structure and their effect on mechanical properties of Al/Mg dissimilar friction stir welding joints[J]. Journal of Materials Science & Technology, 2017, 33(4): 359-366”, they have found the IMCs layer in Al/Mg FSW joint and explained this phenomenon by element diffusion. According to “Zhou L, Li G H, Zhang R X, et al. Microstructure evolution and mechanical properties of friction stir spot welded dissimilar aluminum-copper joint[J]. Journal of Alloys and Compounds, 2019, 775: 372-382”, Zhou et al calculated the Gibbs energy of the IMCs AlCu, Al2Cu and Al4Cu9 and analyzed the IMC layer formation sequence in Al/Cu dissimilar joint. Fig. 7 depicts the IMCs layer in the stir zone and shoulder affect zone in FSW and SFSW joints. It could be seen that the IMCs layer thickness was different due to different heat accumulation, thus we discussed the underlying interplay between welding temperature and the formation of IMCs in Fig 10 and Fig 11 in detail.

It also explains the IMCs layer of SAZ is thicker than that of SZ in both FSW and SFSW joints, as the SAZ is subject to more frictional heat.

 We have added this part in the 243rd line.

  1. Microhardness measurement without indicated error cannot show the difference in values, there may be an error so large that there is no change in microhardness.

RESPONDS:

Thank you for the reminding. We have added the error in the microhardness measurement.

Each point was measured three times.

We have added this part in the 122nd line in Materials and Methods section

The error bar fluctuated within 4 HV, representing the microhardness result of each point was reliable.

We have added this part in the 250th line.

  1. At what strain rate were tensile tests performed?

RESPONDS:

Thank you for the reminding.

Three standard tensile samples were cut perpendicular to the weld for each joint to perform the tensile tests at a constant strain rate of 5 × 10-4 s.

We have added it in the 108th line.

  1. Possible compositions in Table 3 are unproven.

RESPONDS:

Thank you for the reminding. Since the XRD result can identify the IMCs in the fracture surface. Thus, we decided to delete the EDS results and Table 3.

Reviewer 4 Report

The paper can be published.

Author Response

Thank you for the comments concerning our manuscript